# MONOTONIC NEURAL NETWORK: COMBINING DEEP LEARNING WITH DOMAIN KNOWLEDGE FOR CHILLER PLANTS ENERGY OPTIMIZATION

## ABSTRACT

In this paper, we are interested in building a domain knowledge based deep learning framework to solve the chiller plants energy optimization problems. Compared to the hotspot applications of deep learning (e.g. image classification and NLP), it is difficult to collect enormous data for deep network training in real-world physical systems. Most existing methods reduce the complex systems into linear model to facilitate the training on small samples. To tackle the small sample size problem, this paper considers domain knowledge in the structure and loss design of deep network to build a nonlinear model with lower redundancy function space. Specifically, the energy consumption estimation of most chillers can be physically viewed as an input-output monotonic problem. Thus, we can design a Neural Network with monotonic constraints to mimic the physical behavior of the system. We verify the proposed method in a cooling system of a data center, experimental results show the superiority of our framework in energy optimization compared to the existing ones.

## 1 INTRODUCTION

The demand for cooling in data centers, factories, malls, railway stations, airports and other buildings is rapidly increasing, as the global economy develops and the level of informatization improves. According to statistics from the International Energy Agency (IEA, 2018), cooling energy consumption accounts for 20 of the total electricity used in buildings around the world today. Therefore, it is necessary to perform refined management of the cooling system to reduce energy consumption and improve energy utilization. Chiller plants are one of the main energy-consuming equipment of the cooling system. Due to the non-linear relationship between parameters and energy consumption, and performance changes due to time or age, deep learning is very suitable for modeling chiller plants.

In recent years, deep learning (Goodfellow et al., 2016) research has made considerable progress, and algorithms have achieved impressive performance on tasks such as vision (Krizhevsky et al., 2012; He et al., 2016), language (Mikolov et al., 2011; Devlin et al., 2018), and speech (Hinton et al., 2012; Oord et al., 2016), etc. Generally, their success relies on a large amount of labeled data, but real-world physical systems will make data collection limited, expensive, and low-quality due to security constraints, collection costs, and potential failures. Therefore, deep learning applications are extremely difficult to be deployed in real-world systems.

There are some researches about few sample learning summarized from Lu et al. (2020), which focusing on how to apply the knowledge learned in other tasks to few sample tasks, and applications in computer vision, natural language processing, speech and other tasks. Domain Knowledge that has been scientifically demonstrated, however, is more important in few sample learning tasks, especially in the application of physical system optimization.

Domain knowledge can provide more derivable and demonstrable information, which is very helpful for physical system optimization tasks that lack samples. We discussed the method of machine learning algorithms combined with domain knowledge and its application in chiller energy optimization in this article.

In particular, we propose a monotonic neural network (MNN), which can constrain the input-output of the chiller power model to conform to physical laws and provide accurate function space about chiller plants. Using MNN for system identification can help the subsequent optimization step and improve 1.5% the performance of optimization compared with the state-of-the-art methods.

## 2 BACKGROUND AND RELATED WORK

Chiller plants[1] energy optimization is an optimization problem of minimizing energy. In order to simplify the optimization process, the optimized system is usually assumed to be stable, which means that for each input of the system, the corresponding output is assumed to be time-independent. Mostly used methods are model-based optimization (MBO[2]) (Ma & Wang, 2009; Ma et al., 2011; Huang & Zuo, 2014). Although Some research using Reinforcement learning model for optimal control (Wei et al., 2017; Li et al., 2019; Ahn & Park, 2020). However, applying RL to the control of real-world physical systems will be caused by unexpected events, safety constraints, limited observations, and potentially expensive or even catastrophic failures Becomes complicated (Lazic et al., 2018).

MBO has been proven to be a feasible method to improve the operating efficiency of chillers, which uses chiller plants model to estimate the energy consumption with given control parameters under the predicted or measured cooling load and outside weather conditions. The optimization algorithm is then used to get the best value of the control parameter to minimize energy consumption (Malara et al., 2015). The model can be a physics-based model or a machine learning model. Physics-based models are at the heart of today's engineering and science, however, it is hard to apply due to the complexity of the cooling system. Experts need to spend a lot of time modeling based on domain knowledge (Ma et al., 2008). When the system changes (structure adjustment, equipment aging, replacement), it needs to be re-adapted. In recent years, the data-driven method has gradually become an optional solution. Its advantage lies in the self-learning ability based on historical data and the ability to adapt to changes. Thanks to its stability and efficiency, linear regression is the mostly used modeling method in real-world cooling system optimal control tasks (Zhang et al., 2011; Lazic et al., 2018). But ordinary linear models cannot capture nonlinear relationships between parameters and energy consumption, and polynomial regression is very easy to overfit. With the remarkable progress of deep learning research, some studies apply it in cooling system (Gao, 2014; Evans & Gao, 2016; Malara et al., 2015). Deep learning is very good at nonlinear relationship fitting, but it relies on a large amount of data and is highly nonlinear, which brings great difficulties to subsequent decision-making. Due to the inability to obtain a large amount of data, frontier studies have begun to consider the integration of domain knowledge into the progress of system identification and optimization (Vu et al., 2017; Karpatne et al., 2017; Muralidhar et al., 2018; Jia et al., 2020). The combination methods made laudable progress, although it is still at a relatively early stage.

In conclusion, reinforcement learning approach either requires a detailed system model for simulation or an actual system that can be tested repeatedly. The cooling system is too complex to simulate, the former is impossible. While in actual system design and implementation, the latter may be impractical. The MBO method has been proven to be feasible in optimal control, and the optimization performance is determined by the system identification model. However, physical model is complex and time-consuming, linear model in the machine learning model has poor fitting ability, neural network requires large scale datasets, and its highly nonlinearity is not conducive to subsequent optimization step. Domain knowledge can provide more knowledge for machine learning, in this article, we make a theoretical analysis and methodological description about the combination of domain knowledge and deep networks. In particular, we propose a monotonic neural network, which can capture operation logic of chiller. Compared with the above state of art method, MNN reduces the dependence on amount of data, provides a more accurate function space, facilitates subsequent optimization steps and improves optimization performance.

---

[1]How chiller plants work can see in appendix A.1.
[2]How MBO methods work can see in appendix A.3.

## 3 MACHINE LEARNING COMBINE WITH DOMAIN KNOWLEDGE

Consider a general machine learning problem, let us explain the method of machine learning from another angle. It is well known that the life cycle of machine learning modeling includes three key elements: Data, Model, and Optimal Algorithm.

$$f^* = \arg\min_{f \in \mathbb{F}} R_{exp} \text{ s.t. constraints} \tag{1}$$

First, a function representation set is generated through a model. Then Under the information constraints of training datasets, the optimal function approximation is found in the function set through optimization strategies. Deep learning models have strong representation capabilities and a huge function space, which is a double-edged sword. In the case of few sample learning tasks, if we can use domain knowledge to give more precise function space, more clever optimization strategies, and more information injected into the training datasets. Then the function approximation to be solved will have higher accuracy and lower generalization error.

Prior knowledge is relatively abstract and can be roughly summarized as: properties (Relational, range), Logical (constraints), Scientific (physical model, mathematical equation). Several methods of how domain knowledge can help machine learning are summarized in this paper, as follows:

**Scientific provides an accurate collection of functions.** If the physical model is known but the parameters are unknown, machine learning parameter optimization algorithms and training samples can be used to optimally estimate the parameters of the physical model. This reduce the difficulty of modeling physical models.

**Incorporating Prior Domain Knowledge into data.** The machine learning algorithm learns from data, so adding additional properties domain knowledge to the data will increase the upper limit of model performance, such as: Constructing features based on the correlation between properties; processing exceptions based on the legal range of properties; Data enrichment within the security of the system, etc.

**Incorporating Prior Domain Knowledge into optimal algorithm.** The optimization goals in machine learning can be constructed according to performance targets. Therefore, logic constraints in domain knowledge that have an important impact on model performance can be added as a penalty to the optimization objective function. That will make the input and output of the model conform to the laws of physics, and improve the usability of the model in optimization tasks.

**Incorporating Prior Domain Knowledge into model.** Another powerful aspect of deep learning is its flexible model construction capabilities. Using feature ranges and logical constraints of domain knowledge can guide the design of deep learning model structure, which can significantly reduce the search space of function structure and parameters, improve the usability of the model.

## 4 CHILLER PLANTS ENERGY OPTIMIZATION

This section will introduce the application of using the machine learning combine with domain knowledge to optimize the energy consumption of chiller plants. The algorithm model mentioned below has been actually applied to a cooling system of a real data center. We use model-based optimization method to optimize chiller plants. The first step is to identify the chiller plants. We decompose the chiller plants into three type models: cooling/chilled water pump power model, cooling tower power model, and chiller power model , see Equation 2.

$$y = P_{CH} + P_{CT} + P_{COWP} + P_{CHWP} \tag{2}$$

### 4.1 MODEL WITH SCIENTIFIC

For the modeling of the cooling tower power and the cooling/chilled pump power, we know the physical model according to domain knowledge, that is, the input frequency and output power are cubic relationship (Dayarathna et al., 2015), see Equation 3.

$$y = f(x; \boldsymbol{\theta}) = P_{de} \cdot [\theta_3 \cdot (x/F_{de})^3 + \theta_2 \cdot (x/F_{de})^2 + \theta_1 \cdot (x/F_{de}) + \theta_0] \tag{3}$$

Where $x$ is the input parameter: equipment operating frequency; $P_{de}$ is the rated power. $F_{de}$ is the rated frequency, which is a known parameter that needs to be obtained in advance. $\theta_0, \theta_1, \theta_2, \theta_3$ is the model parameter that needs to be learned.

## 4.2 FEATURES WITH PROPERTIES

For the modeling of chiller power, we can integrate the relationship information between properties into the features to improve the fitting ability of the model by analyzing how the chiller plants work in appendix A.1.

$$y_{CH} \propto T_{condenser}, Q_{cooling\_loads} \tag{4a}$$
$$T_{condenser} \propto T_{cow\_in}, F_{cow\_pump} \tag{4b}$$
$$T_{cow\_in} \propto T_{cow\_out}, T_{wb}, 1/F_{fan} \tag{4c}$$
$$T_{cow\_out} \propto T_{condenser}, T_{cow\_in}, 1/F_{cow\_pump} \tag{4d}$$
$$Q_{cooling\_loads} \propto (T_{chw\_in} - T_{chw\_out}), Q_{chilled\_water\_flow} \tag{4e}$$
$$Q_{chilled\_water\_flow} \propto F_{cow\_pump} \tag{4f}$$

See Equation4 lists the causal relations between $y_{CH}$ and the variables on the cooling side and chilled side, and the correlation between variables. Because $T_{cow\_in}$ and $T_{cow\_out}$ is an autoregressive attribute related to time series, so it cannot be used as a feature. We will get features, list in Equation 5.

$$\mathbf{x}_{CH} = (T_{wb}, T_{chw\_out}, T_{chw\_in}, F_{cow\_pump}, F_{fan}, F_{chw\_pump}) \tag{5}$$

## 4.3 OBJECTIVE FUNCTION WITH LOGIC

For the modeling of chiller power, we choose to use MLP as the power estimation model of chiller in the choice of model structure. The MLP model has the advantage to fit well on the nonlinear relationship between input and output. However, the estimated hyperplane of chiller power$(\mathbf{c}, f_{chiller}(\mathbf{x}))$ has the bad characteristics of non-smooth and non-convex due to limited data and the highly nonlinearity of the neural network, resulting in the estimation hyperplane of total power, that will be optimized, $(\mathbf{c}, f_{total}(\mathbf{x}))$ has multiple local minimum points, see figure 4.1 . Moreover, the input and output of the model do not match the operating principle of the chiller from the performance curve. This brings great difficulties to the optimization steps later, which is why deep learning is rarely used in the control of real physical systems.

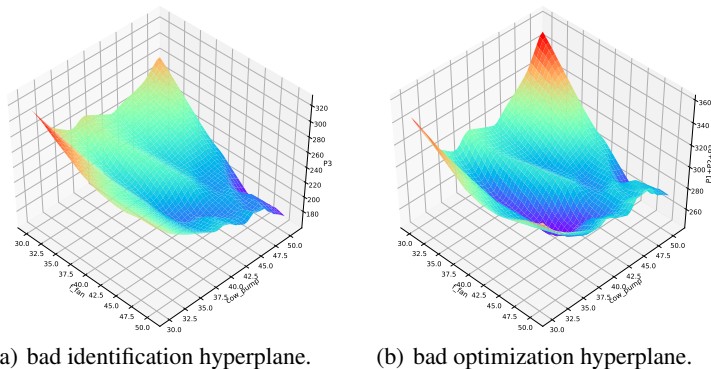

(a) bad identification hyperplane.   (b) bad optimization hyperplane.

Figure 4.1: natural curve.

The chiller plants have the following operating logic, such as the cooling tower fan increases the frequency, and will decrease the power of the chiller, etc. So the model's natural curve [3] of parameters should be monotonous, see Table 1. The natural curve output by the vanilla MLP model does not conform to this rule, see Figure 4.2.

---

[3]The natural curve or called sensitivity curve: the change curve of y along a certain dimension of X.

Table 1: x - $P_{CH}$ monotonicity

| x | Monotonicity |
|---|---|
| $F_{fan}$ | Decrease ↘ |
| $F_{cow\_pump}$ | Decrease ↘ |
| $T_{wb}$ | Increase ↗ |
| $F_{chw\_pump}$ | Increase ↗ |
| $T_{chw\_out}$ | Decrease ↘ |
| $T_{chw\_in}$ | Increase ↗ |

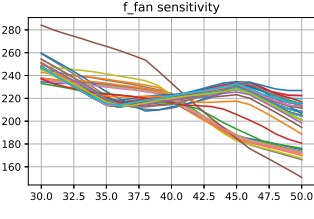

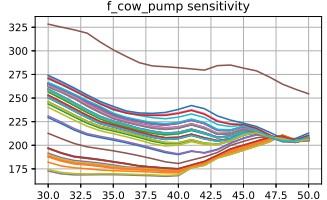

(a) bad natural curve of $F_{fan}$.      (b) bad natural curve of $F_{cow\_pump}$.

Figure 4.2: bad natural curve. Each curve is a sample

Adding a penalty for the inconsistency of the physical law (monotonicity) to the loss function can achieve the effect of incorporating the constraints of the chiller operating logic into the chiller model. Here we design two pairwise rank loss[4] for that:

$$Loss(\hat{y_A}, \hat{y_B})_{rank} = CrossEntropy(Sigmoid(\hat{y_A} - \hat{y_B}), \mathbb{I}(y_A > y_B)) \quad (6)$$

$$Loss(\hat{y_A}, \hat{y_B})_{rank} = max(0, \hat{y_A} - \hat{y_B}) \cdot \mathbb{I}(y_A < y_B) + max(0, \hat{y_B} - \hat{y_A}) \cdot \mathbb{I}(y_A > y_B) \quad (7)$$

In Equation 6, we use the $sigmoid$ function to map the difference between the power estimated label of the A sample and B sample into the probability estimate of $y_A > y_B$, and then use cross entropy to calculate the distance between the estimated probability distribution $Sigmoid(\hat{y_A} - \hat{y_B})$ and the true probability distribution $\mathbb{I}(y_A > y_B)$ as a penalty term.

In Equation 7, when the estimated order of the label of A sample and B sample does not match the truth, we use the difference of the estimated label of the label of A sample and B sample as a penalty.

Based on the addition of the above penalty items, the learning of the model can be constrained by physical laws, so that the natural curve of the model conforms to monotonicity, the effect See Figure 4.3, and the estimated hyperplane is very smooth, and the optimized plane is also convex It is easy to use the convex optimization method to obtain the optimal control parameters. see Figure 4.4.

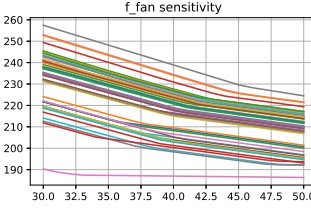

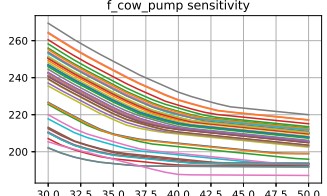

(a) good natural curve of $F_{fan}$.      (b) good natural curve of $F_{cow\_pump}$.

Figure 4.3: good natural curve.

---

[4]$\mathbb{I}$ is Indicator Function

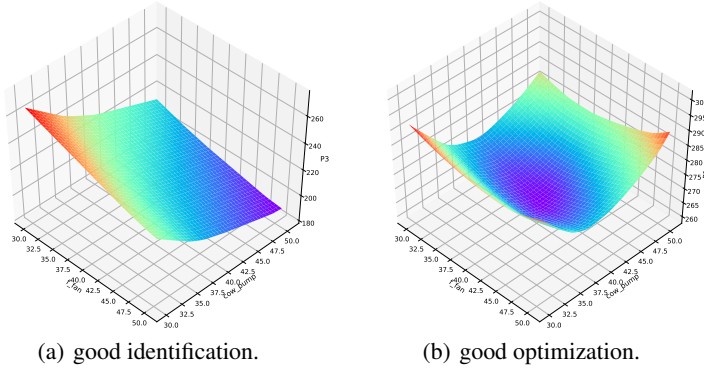

(a) good identification.    (b) good optimization.

Figure 4.4: good identification and optimization hyperplane.

The addition of the rank loss requires us to construct pairwise samples $[(\boldsymbol{x}_A, \boldsymbol{x}_B), \mathbb{I}(y_A > y_B)]$. Part of the construction comes from original samples, and others need to be generated extra. First $\boldsymbol{x}_B$ is copy from $\boldsymbol{x}_A$, then $\boldsymbol{x}_B$ selects a monotonic feature $\boldsymbol{x}_{*_B}$ plus a small random value. Based on the order of $\boldsymbol{x}_{B_*}$ and $\boldsymbol{x}_{A_*}$, referring to the monotonicity of $\boldsymbol{x}_*$ we will get the true power consumption comparison $\mathbb{I}(y_A > y_B)$.

## 4.4 MODEL STRUCTURE WITH LOGIC

The former Section 4.3 describes the integration of logic constraints by adding penalty items to the loss function, so that the trained model conforms to the physical law of monotonic input and output. This section will describe how to use parameter constraints $constraints(\boldsymbol{\theta})$ and model structure design $\dot{f}$ to further improve the model's compliance with physical laws and model accuracy. see Equation 8.

$$y = \dot{f}(\mathbf{x}, constraints(\boldsymbol{\theta})), \text{ s.t. x-y satisfies Physical Law} \tag{8}$$

Inspired by ICNN(Amos et al., 2017), we designed a Monotonicity Neural Network, which gives the model the monotonicity of input and output through parameter constraints and model structure design, called hard-MNN. Corresponding to this is the model in the previous section that learns monotonicity through the objective and loss function called soft-MNN.

### 4.4.1 HARD-MNN

Model structure see Figure 4.5. The main structure of the model is a multi-layer fully connected

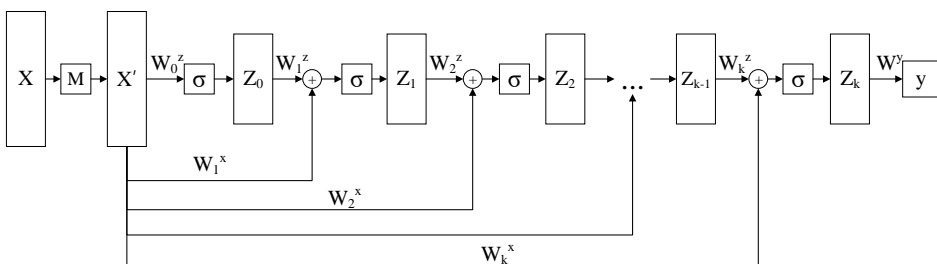

Figure 4.5: hard-MNN. X is Input, y is Output, M is mask layer, $Z_i$ is hidden layer, $W$ is weights: $W^x$ is passthrough layer weights, $W^z$ is main hidden layer weights. $W^y$ is output layer weights, $\sigma$ is activate function, $+$ is aggregation function.

feedforward neural network, and the mask layer function 9 is added after the input layer to identify

the monotonic direction of $x_i$. If $x_i$ decreases monotonously, take the opposite number, otherwise it remains unchanged.

$$f_m(x) = \begin{cases} -x & \text{if } x \in \text{Increase set} \\ x & \text{if } x \in \text{Increase set} \end{cases} \tag{9}$$

In the model definition, we constrain the weight to be non-negative ($W^x \geq 0, W^y \geq 0, W^z \geq 0$). Combined with the mask layer, we can guarantee the physical laws of monotonically increasing or decreasing from the input to the output. Because the non-negative constraints on the weights are detrimental to the model fitting ability, a "pass-through" layer that connects the input layer to the hidden layer is added to the network structure to achieve better representation capabilities. There are generally two ways of aggregate function, plus or concate, which can be selected as appropriate, but the experimental results show that there is no significant difference.

$$z_i = \begin{cases} W_i^{(z)} z_{i-1} + W_i^{(x)} x' & \text{plus} \\ [W_i^{(z)} z_{i-1}; W_i^{(x)} x'] & \text{concate} \end{cases} \tag{10}$$

Similar to common ones are residual networks (He et al., 2016) and densely connected convolutional networks (Huang et al., 2017), the difference is that they are connections between hidden layers. What needs to be considered is that the non-negative constraint of weights is also detrimental to the fitting ability of nonlinearity. It makes the model only have the fitting ability of exponential low-order monotonic functions. Therefore, some improvements have been made in the design of the activation function. Part of the physical system is an exponential monotonic function, but in order to improve the versatility of the model, we designed a parametric truncated rectified Linear Unit (PTRelu)11, which can improve the ability to fit higher-order monotonic functions .

$$f_\sigma(x) = min(\alpha \cdot sigmoid(\beta x), max(0, x)) \tag{11}$$

$\alpha, \beta$ are hyperparameter or as learnable parameters, $\alpha$ is the upper bound value of the output of the activation function, and $\beta$ determines the smoothness of the upper bound to ensure its high-order nonlinearity and weaken the gradient explosion. Input-output comparison of activation function see Figure 4.6

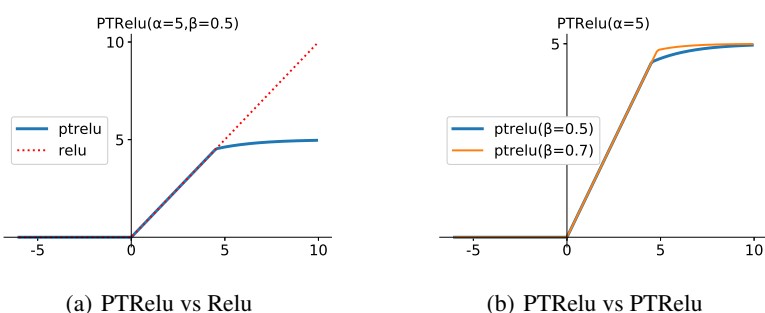

(a) PTRelu vs Relu                    (b) PTRelu vs PTRelu

Figure 4.6: PTRelu.

In addition, we extend the monotonic neural network to make it more general refer to (Amos et al., 2017; Chen et al., 2019). Such as: partial monotonicity neural network in A.4, monotonicity recurrent neural network in A.5 etc.

Adding each power model will get a total power model with convex properties, which is similar to ICNN. However, ICNN only guarantees the convex function properties of the objective function, which can facilitate the optimization solution but does not guarantee the compliance of the physical laws, nor the accuracy of the optimal value.

## 5 EXPERIMENTS

We evaluate the performance of MNN-based and MLP-based optimization methods in a large data center cooling system. Since the performance of MBO mainly depends on the quality of the basic model, we first compare the accuracy of the two system identification models. Then we compare the energy consumption of the two models under the same cooling load and external conditions.

**Comparison of model estimation accuracy.** From figure 5.1 we can know, the accuracy and stability of MNNs is better than MLP, because MNN provides a priori and more accurate function space.

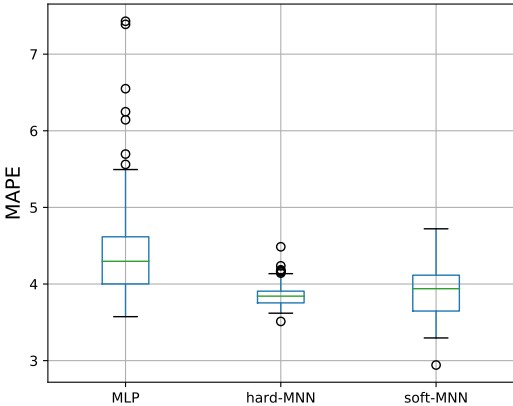

Figure 5.1: Boxplot of mape of MLP, hard-MNN and soft-MNN, which trained on real data collected from a cooling system of a DC. Each model has the same number of hidden layers and the number of neurons in each layer, as well as the same training set, test set, and features. The result is obtained after 100 non-repetitive tests.

**Comparison of energy consumption.** Considering that energy consumption is not only related to interlnal control but also related to the external conditions (cooling load and outside weather), in order to ensure the rationality of the comparison, we make PUE comparisons at the same wet bulb temperature. As shown in figure 5.2, hard-MNN is more energy-efficient, stable and finally reduces the average PUE by about 1.5% than MLP.

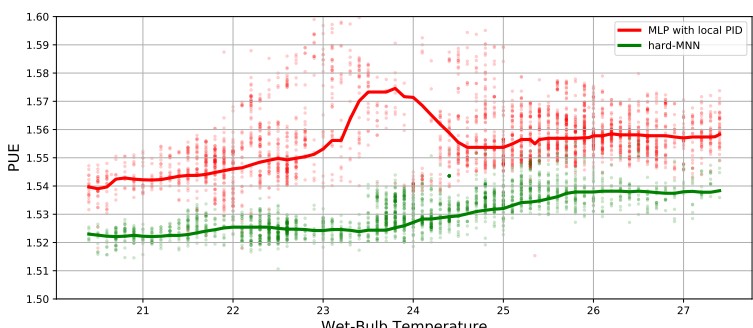

Figure 5.2: Energy consumption comparsion in real system. MLP is hard to be used in real world system optimization due to highly nonlinear, so we use mlp with local PID for safe constraints.

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

# A  APPENDIX

## A.1  COOLING SYSTEM

As shown in Figure A.1, chiller plants are the main equipment of the cooling system. The chiller is used to produce chilled water. The chilled water pump drives the chilled water to flow in the water pipe and distributes it to the air handling units (AHUs). The fan of AHUs drives the cold air to exchange heat with the indoor hot air for cooling rooms. In this process, the heat obtained by the chiller from the chilled water needs to be dissipated into the air through equipment such as cooling towers. Most of the heat exchange process uses water as a medium, and the equipment that drives the flow of the medium is cooling water pump. Chillers, water pumps, fans of AHUs and fans of cooling towers constitute the main components of the energy consumption of the cooling system. For more details, please refer to Stanford III (2011)

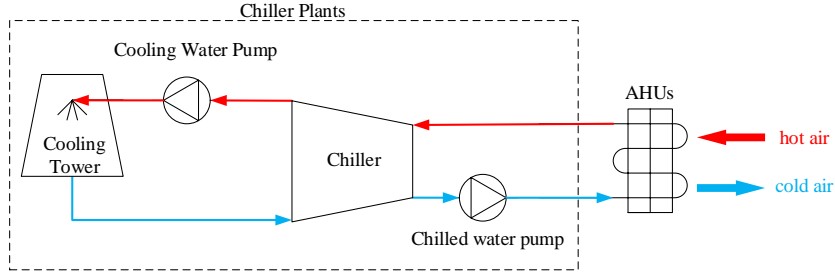

Figure A.1: Cooling System Structure.

Table 2: Table of notations

| Symbol | Description |
|---|---|
| $\mathbf{c}$ | Control vector-variables |
| $\mathbf{s}$ | State vector-variables |
| $\mathbf{x}$ | Features of model, contains $\mathbf{c}$ and $\mathbf{s}$ |
| $y$ | Totoal power of chillers, cooling towers and water pumps |
| $\boldsymbol{\theta}$ | Parameters of identification model |
| $F_{cow\_pump}$ | Frequency of cooling water pump |
| $F_{fan}$ | Frequency of cooling tower fan |
| $T_{wb}$ | Temperature of Wet bulb |
| $T_{chw\_out}$ | Temperature of chilled water flow out chillers |
| $T_{chw\_in}$ | Temperature of chilled water flow in chillers |
| $F_{chw\_pump}$ | Frequency of cooling water pump |
| $T_{cow\_out}$ | Temperature of cooling water flow out chillers |
| $T_{cow\_in}$ | Temperature of cooling water flow in chillers |
| $P_{CH}$ | power of chillers |
| $P_{CT}$ | power of cooling towers |
| $P_{COWP}$ | power of cooling water pumps |
| $P_{CHWP}$ | power of chilled water pumps |

## A.2  NOTATION

We have summarized the symbols used in the article, see Tabel 2. There are two types of variables for data collection in the cooling system: control variables $\mathbf{c}$ and state variables $\mathbf{s}$ and powers. Control variables are parameters that can be manually adjusted, state variables are factors that are not subject to manual adjustment, but they all affect the energy consumption of the system. $\mathbf{x}$ is the input feature of models and $y$ is the output target of models. $\boldsymbol{\theta}$ represents the parameters of models. The symbols below represent actual variables in the cooling system. $F_{cow\_pump}$, $F_{fan}$ are the control variables we want to optimize. $T_{wb}, T_{chw\_out}, T_{chw\_in}, F_{chw\_pump}, T_{cow\_out}, T_{cow\_in}$ are environment variables[5]. $P_{CH}, P_{CT}, P_{COWP}, P_{CHWP}$ are the power of each equipment in chiller plants.

## A.3  OPTIMAL CONTROL

Chiller plants energy optimization is an optimization problem of minimizing energy. In order to simplify the optimization process, the optimized system is usually assumed to be stable, which means that for each input of the system, the corresponding output is assumed to be time-independent. Commonly used methods are model-free strategy optimization or model-based optimization. The strategy optimization method is to control according to the rules summarized by experience. The

---

[5]$T_{chw\_out}, F_{chw\_pump}$ can also be controlled, but they will affect the energy consumption of AHUs. So in order to simplify the optimization process, no optimization control is performed on them.

model-based optimization method has two steps, including system identification and optimization, see Figure A.2.

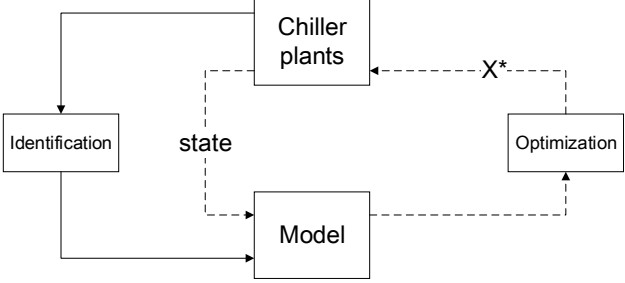

Figure A.2: Mobel based optimal control. Solid line is identification step, dotted line is optimization step.

The first step is to model the system, that is, building mapping function $f : \boldsymbol{x} \rightarrow y$ between features and energy consumption as shown in Equaltion 12, this step is usually done offline. In the second step, a constrained objective function is created based on the function of the first step, and then use the optimization algorithm to find the optimal value of the control parameter.The solved values will be sent to the controller of the cooling system, this step is usually performed online.

$$
\begin{aligned}
&1.identification: \\
&\qquad y = f(\boldsymbol{x}; \boldsymbol{\theta}) \\
&2.optimization: \\
&\qquad \mathbf{x}^* = \arg\min_{\mathbf{x} \in \mathbb{X}} f(\boldsymbol{x}; \boldsymbol{\theta}), \text{ s.t. some constraints}
\end{aligned}
\tag{12}
$$

The modeling in the first step is the key step and the core content of this article, because it directly determines whether the implementation of optimization is troublesome, and indirectly determines the accuracy of the optimal value.

## A.4   PARTIAL-MNN

Of course, when applied to other scenarios, the structure of hard-MNN is not applicable because the features may not conform to all x-y monotonicity, so we expand hard-mnn to partial-mnn, and the model structure see Figure A.3. The partial-MNN has one more branch network parts compared with hard-MNN, and the mask layer has also been modified.

The partial mask layer, see Equation 13 is designed to identify monotonic decreasing, monotonic increasing and non-monotonic features.

$$
f_{m1}(x) = \begin{cases} 0 & \text{non-Monotonic} \\ -x & \text{Decrease} \\ x & \text{Increase} \end{cases}
\tag{13a}
$$

$$
f_{m2}(x) = \begin{cases} x & f_{m1}(x) = 0 \\ 0 & f_{m1}(x) \neq 0 \end{cases}
\tag{13b}
$$

The monotonic feature is input into the backbone network through the mapping of $f_{m1}$ of the mask layer, $\mathbf{x}_m = f_{m1}(\mathbf{x})$. Non-monotonic features are input into the branch network $\mathbf{x}_n = f_{m2}(f_{m1}(\mathbf{x}))$ through the $f_{m2}$ mapping of the mask layer.

The branch network has no parameter constraints, uses the ordinary relu activation function, and merges with the backbone network at each layer, see Figure A.3

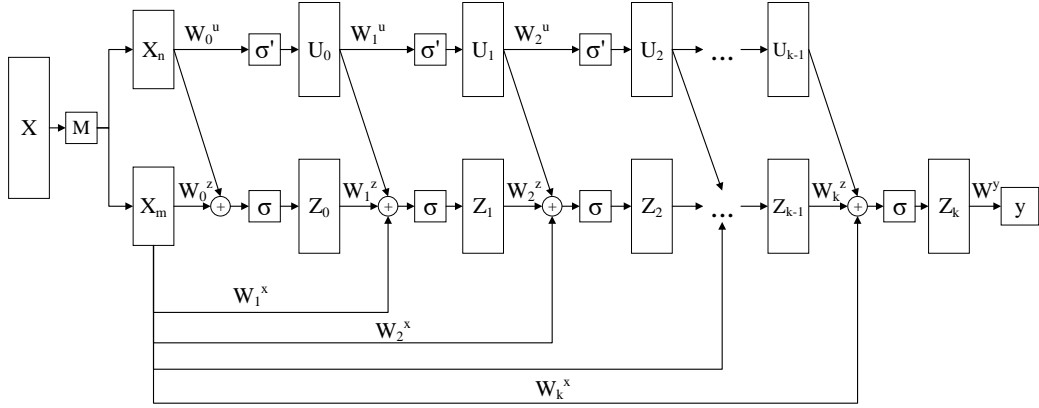

Figure A.3: partial-MNN.

## A.5 MRNN

MRNN replaces the main structure with RNN to support the modeling of timing-dependent systems, and increases the monotonicity of the timing dimension by constraining parameters compared to MNN. As we mentioned earlier, the cooling system is a dynamic system with time delay. In order to simplify the system, it is assumed that the system is a non-dynamic system. When the collected data granularity is dense enough, MRNN can be used to model the chiller plants. MRNN model structure see Figure A.4. In the model structure, we constrain part of the weight parameters to be non-negative (st $U \geq 0, V \geq 0, W \geq 0, D_1 \geq 0, D_2 \geq 0, D_3 \geq 0$) to ensure the monotonicity of input and output The performance and timing are monotonic, and a mask layer is added to the input layer. Use the Ptrelu activation function, and the output layer is Relu. $D_1, D_2, D_3$ are the weights of the pass through layer to improve the fitting ability of the network.

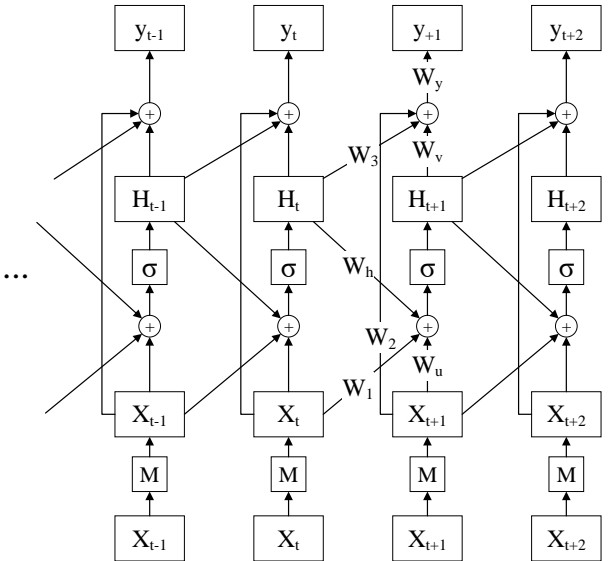

Figure A.4: MRNN.

