# OpenReview forum: "Monotonic neural network: combining deep learning with domain knowledge for chiller plants energy optimization"
_ICLR.cc/2021/Conference — Reject_

### Official Review · AnonReviewer1 · 2020-10-28
**The manuscript has structural issues along with inadequate justification on methods proposed**

**Rating:** 3
**Confidence:** 4

**Review:**

Many thanks to the authors for their submission which aims at combining data-driven approaches with model-based ones for optimising centralised cooling systems (chiller plants). The overarching aim of this paper has been well studied before; given one aims at incorporating physical constraints, one needs a solid understanding of the underlying physical system, hence how one adds such a constraint to the model requires deep investigations. The machine learning part of this paper is also not something new. To match the physical constraints of the problem in hand authors incorporate monotonicity to the Neural Network model. Other than that, the paper does not offer a comprehensive explanation on what the novelty and actual contributions are, nor does it provide a solid experimental set-up. Please consider some of my comments below:

Major comments:
--No conclusion or discussion has been provided. That makes the paper incomplete
--Experimental section is rather inadequate and rushed up; the results are not explained properly; no concrete information on the dataset used has been provided to contextualise the experimental design.
--Paper is heavily skewed towards providing too much background information and rather simplistic information on incorporating physical constraints on to the loss function of a neural-net.

Minor comments: Quite a few punctuation errors exist. Some sentences are not finished or finish with a full stop where it shouldn't. Please consider proof-reading the manuscript carefully for correcting such typos.

---

### Official Review · AnonReviewer4 · 2020-10-28
**Presenting a method of utilizing domain knowledge for neural network design and applying it to chiller plant optimization**

**Rating:** 2
**Confidence:** 4

**Review:**

This paper presents how to utilize the domain knowledge which might be useful for solving problems in developing a deep neural network with the forms of variable constraints or objective functions. And the proposed method, called 'Monotonic Neural Network', was applied to the application of chiller plant optimization. Compared to the method (MLP), the proposed method showed better performance in the accuracy measure.

Though this article shows how the machine learning model can be guided with prior knowledge in the real-world application, it is hard to say that this paper is above the accept line as there are lots of rooms for improvement.

Completeness
- No conclusion: To make the research more consolidate, authors need to consider putting conclusions of their works including summary, reemphasizing what they found, some key points, or future works. By just finishing with 'Experiments' in this article, readers may wonder what are the main points of arguments from authors.
- Explanations for abbreviation: As the paper deals with the application of  'Chiller plant', using technical terms is inevitable(e.g PUE, We-Bulb Temperature, DC, or CHWP). However, authors need to bring the meaning of the terms from the appendix into the main article so that readers can glimpse easily when the unknown terms are shown.

Applicability
- Its applicability is limited only to the same problem from the same facilities. Authors may consider making their proposed method more general so that practitioners from other industries who want to combine their domain knowledge into their problem can get ways or insights from this study.
- No specific information about the proposed model structure found. How many layers(k) are used and why? What is the size of the layer weights? How are hyperparameters set from how many candidates?

Superiority
- Authors may compare their works to other state-of-the-art approaches to present the superiority of them. Results showing better than MLP may not appeal to readers.

Typo/correction
- Authors need to check the correctness of sentences. Especially, the first paragraph of Section 2 has several misuses of capital letters. And the fifth line(Although Some... ) is not a complete sentence.
- In section 3, Consider -> Considering
- In section 2(Third paragraph), state of art -> state of the art

---

### Official Review · AnonReviewer2 · 2020-10-28
**Monotonic Neural Network: Combining Deep Learning with Domain Knowledge for Chiller Plants Energy Optimization  Score: 4 if by NeurIPS scores (1,2, …, 10), 3 if by ICLR20 scores (1,3,6,8).  Confidence: 4**

**Rating:** 3
**Confidence:** 5

**Review:**

Summary:

This paper tackles the energy optimization problem of chiller plants, which are the main equipment of a cooling system. This paper aims to learn the non-linear mapping between the control parameters (of a cooling system) and energy consumption. The key idea of this paper is to incorporate domain knowledge into the training of neural networks. Specifically, monotonic constraints are proposed to constrain the learning of the non-linear mapping so that physical laws are satisfied. The authors propose two methods to implement monotonicity: hard-MNN and soft-MNN.

Hard-MNN treats monotonicity as hard constraints and uses a Monotonic Neural Network to learn the non-linear mapping. Soft-MNN treats monotonicity as soft constraints and adds a monotonic loss into the training objective to encourage the learned mapping to be monotonic.  Experiments show that (1) both the hard-MNN and soft-MM outperforms multilayer perceptron (MLP), and (2) hard-MNN yields lower energy consumption than MLP.

Pros:

+ The idea of incorporating domain knowledge into training neural nets is reasonable.  (But this is not a new idea.   E.g.,
KG-GAN: Knowledge-Guided Generative Adversarial Networks.)

+ Appendix A.1 does a great job of giving an introduction about cooling systems and chiller plants.

Cons:

- Some claims are unjustified.

Section 2 states that “we make a theoretical analysis and …” However, I cannot find such analysis. Does the theoretical analysis mean Equation 3, 4, and Table 1?

Section 2 states that “Compared with the above state of art method, MNN reduces the dependence on the amount of data, provides a more accurate function space, facilitates subsequent optimization steps and improves optimization performance.” However, the current experimental results are not strong enough to support these four claims.

Specifically, no experiments empirically compare performance in terms of different amounts of training data. There is no attempt to quantify the so-called “a more accurate function space.” No results show information about the optimization applied to the learned mapping function.

- The technical method is not clearly described.

Appendix A.3 gives an overview of how the energy optimization problem is decoupled into two sub-problems: identification and optimization.

I appreciate the overview but also think there is room for improvement.

This part could be important since it gives a concrete high-level view of the problem formulation. In particular, this paper concentrates on the first sub-problem. However, it is not evident to see this point neither from the introduction nor from the beginning of the method section, making it hard for me to understand the problem formulation clearly in my first-pass reading.

In Section 4, three type models are introduced. However, their names and notations are not linked together, which makes the reading very difficult. It would be better if the notations are shown together with their corresponding names like cooling/chilled water pump power model (P_{COWP} and P_{CHWP}), cooling tower power model(P_{CT}), and chiller power model (P_{CH}).

The connections between math equations are loose, which makes the technical method hard to understand. For example,

(1) the connection between y in Equation 2 and y_{CH} in Equation 4 is not clear.
(2) The connection between y_{CH} in Equation 4 and x_{CH} in Equation 5 is also not clear enough.
(3) (c,f_{chiller}(x)) mentioned in Section 4.3 is somewhat isolated, its relationship with the other equations are not evident.

The last five lines of Section 4.3 (below Figure 4.4) are hard to understand. Overall, the technical writing of Section 4 could be further improved.

Figure 4.1 and 4.4 lack a detailed explanation and thus are hard to understand. What do f_fan, cow_pump, P3, P1+P2+P3 in the figure mean, respectively? What is the difference between (a) and (b)? Why (a) is called identification hyperplane, while (b) is called optimization hyperplane?

Most of the math notations are not explicitly explained in the paragraph. I had difficulty reading the mathematical equations until I found the table of notations in the Appendix (Table 2). However, Table 2 is not referenced throughout the paper.

- Insufficient experiments in terms of setting, details, and comparisons.

The Experiments section lacks a detailed description of dataset information, training settings, implementation details. The current experiments look impossible to reproduce the results. What is the general introduction of the dataset used in the experiments? Do you split the data into training, validation, and testing set? What is the number of data examples of each subset? How to train the proposed model? What is the detailed network architecture of the MLP and the proposed MNN?

In the first experiment (Figure 5.1), what is MAPE, and how to calculate it? What is DC mentioned in Figure 5.1?
The second experiment (Figure 5.2) is unclear in its purpose, comparison method, metric, and effectiveness.
It is hard to judge the effectiveness of the proposed method based on the experiments.
Why do we need to consider experimenting with different wet-bulb temperatures?

What is the key difference in terms of the experiment setting compared with the previous experiment? The 1st experiment focuses on internal control, while the 2nd experiment focuses on external control?
Can the authors elaborate more on these two experiments (Figure 5.1 and 5.2)?
What is PUE, and how to calculate it?
What do the dots and lines in Figure 5.2 mean, respectively?
What is MLP with local PID? PID is not explained throughout the paper.
Is improving the average PUE by 1.5% significant?
Why no results of soft-MNN in this experiment?

As the related work mentioned, “linear regression is the mostly used modeling method in real-world …,” a baseline comparison with linear regression looks essential to let readers know the baseline performance. However, the current manuscript compares with MLP only. Is there any reason that prevents from comparing with linear regression?

This paper proposes a new activation function (Equation 11) and two extensions of the proposed monotonic neural networks (MNN): partial-MNN and recurrent-MNN. However, no results are shown to demonstrate their effectiveness.

---

### Official Review · AnonReviewer3 · 2020-10-30
**Monotonic NNs and NNs for chilling plants are old ideas**

**Rating:** 4
**Confidence:** 4

**Review:**

The paper uses monotonic neural networks to estimate the energy consumption of chilling plants. This can be used to save energy by using the neural network in Model-Based Optimization.

On the positive side, the paper actually tests a monotonic neural network in MBO on a real chilling plant, dropping the PUE from 2% (from 1.54 to 1.52, e.g.).

On the negative side, all of the fundamental ideas in the submission are quite old:
*  The idea of penalizing non-monotonic behavior in neural networks was first published in [1].
*  The idea of forcing a monotonic structure on a neural network was first published in [2]. The submission references [3], which enforces convexity (not monotonicity).
*  The idea of using a neural network to predict the energy consumption of a chilling plant was published in [4], although may predate that.

Now the submission does not exactly reproduce these older ideas: e.g., the monotonicity penalty in [1] is quadratic, while in the submission it's either linear, or a log loss mapped through a sigmoid. However, no comparison between these ideas was tested, so it is unclear whether the difference from the old work is significant.

This paper would be of interest to a small segment of engineers who are interested in optimizing chilling plants. It could be made of wider interest if either (a) some stronger mathematical statements could be made, or (b) if more than one empirical experiment were carried out, perhaps across different types of systems. Perhaps the authors could try a system identification benchmark, e.g., [5]?

REFERENCES
1. Sill, Joseph, and Yaser S. Abu-Mostafa. "Monotonicity hints." In Advances in neural information processing systems, pp. 634-640. 1997.
2. Sill, Joseph. "Monotonic networks." In Advances in neural information processing systems, pp. 661-667. 1998.
3. Amos, Brandon, Lei Xu, and J. Zico Kolter. "Input convex neural networks." In International Conference on Machine Learning, pp. 146-155. 2017.
4. Gao, Jim. "Machine learning applications for data center optimization." (2014).
5. nonlinearbenchmark.org

---

### Decision · Program_Chairs · 2021-01-07
**Final Decision**

**Decision:**

Reject

**Comment:**

Thank you for your submission to ICLR.  The reviewers unanimously felt that there were substantial issues with this work, owing to the fact that both the techniques and applications have been considered in a great deal of previous work.  Furthermore, the manuscript itself needs substantial amounts of revision before being suitable for publication.  As there was no response to these points during the rebuttal period, it seems clear that the paper can't be accepted in its current form.